# Coping strategies and psychological distress among mothers during COVID-19 pandemic: The moderating role of social support

**Fitriani Yustikasari Lubis**[1,2]*, **Fitri Ariyanti Abidin**[1,3], **Laila Qodariah**[1,3], **Vidya Anindhita**[1,3], **Fredrick Dermawan Purba**[1,2]

**1** Faculty of Psychology, Universitas Padjadjaran, Bandung, Indonesia, **2** Center for Psychological Innovation and Research, Faculty of Psychology, Universitas Padjadjaran, Bandung, Indonesia, **3** Center for Family Life and Parenting Studies, Faculty of Psychology, Universitas Padjadjaran, Bandung, Indonesia

* fitriani.y.lubis@unpad.ac.id

**Data Availability Statement:** All relevant data are within the paper and its Supporting Information files.

## Abstract

The Indonesian government implemented a large-scale social restriction policy as part of the efforts to tackle the COVID-19 pandemic. This policy impacted the population, including mothers, and caused considerable psychological distress. Individual efforts to cope (avoidant and approach coping strategies) and support from significant persons might help handle the distress experienced by mothers. The purpose of this empirical study is to investigate the effect of individual coping strategies on psychological distress and the moderating role of social support among Indonesian mothers. An online survey was administered from 20th to 25th April 2020 to 1534 Indonesian mothers (Mean age 37.12 years; SD 6.63). Brief COPE (28 items), Depression Anxiety Stress Scale/DASS (18 items), and the Multidimensional Scale of Perceived Social Support/MSPSS (12 items) were used to measure coping strategies, psychological distress, and social support, respectively. IBM SPSS 24 software was used to analyze the data. The result showed that moderate and high levels of social support moderated the relationship between approach coping strategies and psychological distress (B = .041, CI .007-.075). When the mother uses approach coping, her psychological distress will decrease further whenever she receives moderate and high level social support. Any level of social support moderated the relationship between avoidant coping and psychological distress (B = -.100, CI -.138—.061). When mother used avoidant coping, her social support at any level served as buffer to her psychological distress. It can be concluded that mothers need to prioritize implementing approach coping strategies to lower their distress. Those who practiced avoidant coping strategies needed social support from their significant persons to decrease their distress.

## Introduction

The mental health status in various countries is greatly impacted by the outbreak of the coronavirus disease 2019 (COVID-19) [1]. Government regulations, including lockdown and

**Funding:** The authors received no specific funding for this work.

**Competing interests:** The authors have declared that no competing interests exist.

quarantine policies, had an adverse effect on mental health of adults around the world as evidenced by an increase in depression and stress-related symptoms [2, 3], increased parental burnout [4], and the level of parent stress that have not returned to pre-COVID-19 levels [5]. Among others, parents have experienced enormous stress by this pandemic. The multifaceted disruptions in various aspects, such as employment, financial stability, children's education, and childcare responsibilities, collectively gave rise to a profound stressor for parents [6]. A preceding investigation indicated that parents encountered elevated stress levels across several areas of their lives, including work (e.g., unemployment, adaptation to remote work arrangements), familial dynamics (e.g., challenges of remote education, the inability to convene in person), relational intricacies (e.g., marital conflicts and limited support), physical well-being (e.g., inadequate sleep), and psychological well-being (e.g., anxiety, depression) [7]. Another study found that parental mental health issues were associated with low material assets (e.g., financial security, food security, children having access to outdoor space), familial assets (e.g., parents' time for themselves), and community assets (e.g., receiving support from others outside the household) in a pandemic [8]. Consequently, it was more challenging to strike a balance between work with parenting during circuit-breakers [9]. The demands were increased for parents of children with developmental delays, chronic emotional or behavioral difficulties, or other health challenges [10].

Among these effects of the COVID-19 pandemic on parenting, a specific and more profound influence had been found in maternal roles [11, 12]. These effects can be exacerbated in working mothers. According to a previous study, co-parenting had a affected men and women differently during the pandemic [4]. Furthermore, working mothers spent more time in child-rearing and homeschooling than working fathers during this period [13]. A previous study reported that work arrangements during the pandemic resulted in an unideal merger of work and personal space of working mothers [14].

Based on government regulations regarding large-scale social restrictions in Indonesia, schools have been closed since March 2020 until July 2021. Consequently, learning activities have been changed to online activities, leading to changes in Indonesian mothers' behavior and daily lives. Most mothers reported difficulties dividing their time between nurturing children and working [7, 15]. With the center of life happening at home, the role of mothers became more burdensome. Indonesian mothers are expected to bear the responsibility of childrearing and children's school activities while carrying out domestic chores and taking care of all family members [16, 17].

Exposure to stressors, such as those caused by the pandemic, might cause fatigue and furthermore increase strain on the relationship between parents and their children [18]. The study found that the connection between stressful life events and adverse outcomes (e.g., poor parenting behaviors) influenced how potential stressful situations affected adaptive versus maladaptive functioning [19].

Caregiver well-being is crucial for promoting healthy parenting practices and positive marital relations [20, 21], serving as a funnel through which social disruptions due to COVID-19 can infiltrate family functioning via changes to marital, parent-child, and sibling relations that are influenced by caregiver well-being [22]. Managing stress responses to achieve the desired well-being has not only helped the vulnerable to survive this pandemic, but also improved how they would survive an even more challenging life afterward [23], specifically for mothers.

With an important role as the main caregivers in Indonesia's families, mothers in Indonesia should be able to attain their physical and psychological well-being and manage the stress in parenting due to the COVID-19 pandemic. Therefore, it is important to understand the psychological distress condition of mothers and explore the strategies that may promote their physical and psychological well-being.

## Social support

Perceived control is crucial in understanding stress and coping [24]. Coping has been associated with decreases in stress, anxiety, depression, and better adjustment across situations [25–27]. The use of coping strategies has been found to affect and predict perceived control in facing stressors [28]. Coping strategies such as accepting negative thoughts or experiences without judging them are strongly and negatively associated with perceived stress [29]. The types of coping strategies may also interact differently with stressors. Avoidant coping and religious-focused coping emerged as significant predictors of psychological distress [30]. These types of coping also have positive relationships with depression, anxiety, and stress. Meanwhile, self-efficacy, active-practical coping, and active-distractive coping had negative relationships with psychological distress [31]. This findings also occurred during COVID-19, where coping self-efficacy was associated with lower psychological stress [32].

In dealing with the familial and parental stress caused by the pandemic, besides promoting positive coping strategies, it is also important to enhance social support. Several studies have found the role of social support in psychological distress. Social support, self-efficacy, active-practical coping, and active-distractive coping are negatively associated with psychological distress [33]. Amidst the COVID-19 pandemic, active/adaptive coping strategies and increased social support are significantly correlated with decreased psychological distress [1]. Adaptive coping strategies and supportive family environments may serve as protective factors for families experiencing stress [7]. Perceived social support is negatively correlated with anxiety, depression, and stress, indicating that more social support is associated with less depression and anxiety [30, 31, 33, 34]. Previous studies showed that mothers' perceptions of family support were associated with less parenting stress, meaning parents who received more support were better able to engage in positive parenting [35, 36]. Another study on mediation analysis also showed that self-efficacy mediated the effect of social support on depression, anxiety, and stress [33].

## Relationship between social support, coping strategies, and psychological distress

Several studies consistently showed correlations between social support, coping strategies, and psychological distress. For instance, a previous study showed that coping strategies mediated the relationships between perceived social support and anxiety and depression. That study also stated that more facilitative coping (e.g., seeking help) was linked to less anxiety, while more avoidant coping (e.g., avoiding emotions) was linked to more anxiety and depression [34].

Correspondingly, the mediation analyses study reported that avoidance-focused coping mediated the link between social support and depression and between significant other support and anxiety [33]. These dynamics were explained by Budge, S. L., Rossman, H. K., & Howard, K. A. S. (2014) where people with higher social support engage in more facilitative coping, which was correlated with less anxiety, and those who reported less social support engage in more avoidant coping which was correlated with more anxiety [31].

## Current study

Previous studies have demonstrated coping strategies as a mediating variable in the relationship between social support and psychological distress. However, evidence showing social support as a moderating variable between coping strategies and psychological distress is scarce. The present study attempted to seek the moderating role of social support among Indonesian mothers in coping strategies to deal with psychological distress during the COVID-19 pandemic (see Fig 1).

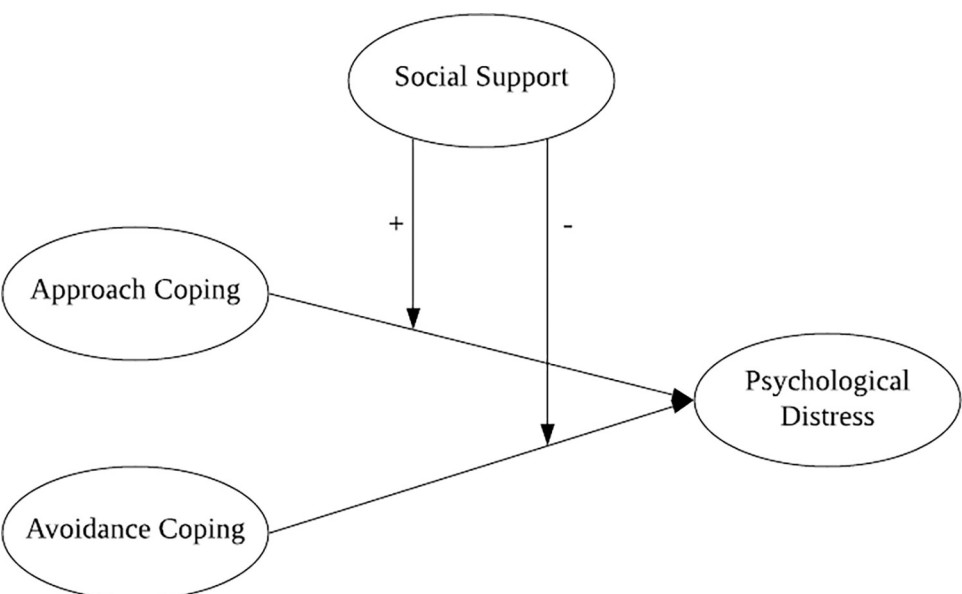

**Fig 1. Conceptual framework of relationship between coping, social support, and psychological distress.**

We hypothesized that:

1. Social support will moderate the relationship between approach coping strategies and psychological distress; the negative relation between approach coping strategies and psychological distress will be stronger for mothers with higher levels of social support.

2. Social support will moderate the relationship between avoidance coping strategies and psychological distress; the positive relation between avoidance coping strategies and psychological distress will be weaker for mothers with higher levels of social support.

## Methods

The present study is part of a longitudinal project investigating the psychological distress of Indonesian mothers and its determinant factors during the COVID-19 pandemic. Only data from the first data collection time point was presented, using a cross-sectional approach. The moderating role of social support was investigated in the relation between individual coping strategies and psychological distress among mothers.

### Participants

The population of this study was Indonesian mothers with children who had to follow large-scale social restrictions implemented by the local and national governments. Inclusion criteria were (i) female, (ii) at least 18 years and older, (iii) has at least one child aged younger than 18 years, (iv) the children lived in the same address as mothers. A convenience sampling method was used to obtain data from the population above.

### Procedures

Data collection was conducted from April 20th to 25th, 2021. The study team shared the Google Form survey link through their network (e.g., parent community at schools, neighborhood community), instant messenger applications (e.g., Whatsapp, Line), and social media (e.g., Instagram,

Facebook). Informed consent was presented and sent through a link to participants. Participants clicked "agree" on the form when willing to participate, otherwise "disagree" was clicked. For participants who are willing to participate, the form would automatically enter the next section, which consist of three measurement tools. Meanwhile, for participants who were not willing to participate, the form will go directly to the closing section. Ethical clearance was obtained from Universitas Padjadjaran Ethical Committee (No. 486/UN6.KEP/EC/2020).

## Measures

A total of four questionnaires were used, namely demographic, Brief COPE, Depression Anxiety Stress Scale/DASS, and the Multidimensional Scale of Perceived Social Support/MSPSS.

Demographic data collected were personal information (i.e., age, education, length of the marriage, number of children, location of residence), information on the spouse (i.e., age, education), and pandemic-related characteristics (i.e., working situation, having a housekeeper, knowing someone in the neighborhood who tested positive for COVID-19).

**Coping strategy.**   Coping strategies were measured using Brief COPE which is a shorter version of the COPE (Coping Orientation to Problem Experienced) inventory [37]. The instrument consists of a 28-item self-report inventory that includes statements indicating the specific strategies for individuals. It measures 14 coping strategies divided into two categories, namely approach coping ("I have been concentrating my efforts on doing something about the situation I am in") and avoidance coping ("I have been criticizing myself") [38]. A total of 14 items indicate each category and participants responded using a Likert scale from 1 for "I have not been doing this at all" to 4 for "I have been doing this a lot" to indicate how frequently the specific coping strategy are used [39]. Higher scores showed the increased practice of the specific coping strategies. The internal consistency in this study was .853 and .723 for approach and avoidance coping, respectively.

**Perceived social support.**   Perceived social support was measured by the Multidimensional Scale of Perceived Social Support (MSPSS), which is a 14-item self-report measure drafted by Zimet et al. (1988) [40]. Items include statements regarding parties who may provide social support to the individual (i.e., family, friends, and significant others), such as "I can talk about my problem with my friends". Participants responded using a Likert scale from 1 for "strongly disagree" to 7 for "strongly agree" to indicate their level of agreement toward the social support received [40]. This scale has been psychometrically validated in the Indonesian language [41]. The higher score indicated higher social support. The internal consistency of the MSPSS in the present sample was .88, .87, and .85 for family, friends, and significant others, respectively.

**Psychological distress.**   Mothers' psychological distress is measured by the Depression Anxiety Stress Scale (DASS), which is an 18-item self-report inventory aimed to measure the presence and severity of symptoms of depression (7 items), anxiety (7 items), and stress (4 items) with a recall period of the previous seven days. An example of the DASS-18 item is "I found it difficult to relax". The response options ranged from 1 for "Did not apply to me at all" to 3 for "Applied to me very much or most of the time". Furthermore, DASS-18 has been validated for the Indonesian population [42]. The higher the score, the more severe the psychological distress. The internal consistency of the domains in the present sample were .82, .75, and .75 for the depression, anxiety, and stress domain, respectively.

## Data analysis

Descriptive data were presented in mean and standard deviations of variables. Pearson's correlational analyses were conducted to examine bivariate relationships between psychological

distress, social support, and coping strategies. After testing the assumptions, this data met the minimum criteria for conducting a moderating analysis using regression (normal distribution, Linearity, Homoscedasticity, Multicollinearity) [43]. For testing the potential moderator, moderated regression model was used through PROCESS analysis (v.3) with 5000 bootstrap samples and heteroscedasticity-consistent standard errors [44]. PROCESS is a conditional process modeling program that makes use of an ordinary least squares or logistic-based path analytical framework to test the moderation effect [45]. Social support was used as the moderator variable (W), psychological distress as the independent variable (X), and approach coping strategies (Y1) and avoidance coping strategies (Y2) as the dependent variables. A total of two separate models were examined to predict that (1) Social support increases the negative association between approach coping and psychological distress and (2) Social support decreases the positive association between avoidance coping and psychological distress. The moderator effect for the two-interaction term indicates a significant moderator effect $p < .05$ and 95% of the level confidence interval (CI). IBM SPSS 24 software was used to analyze the data.

## Results

A total of 136 were excluded from 1670 participants who completed the survey due to invalid answers, aged under 18, and had no child. The final sample was 1534 participants with a mean age of 37.1 years ($SD$ = 6.6). The majority of participants completed their university-level education (84.4%), had a spouse with a university degree (81.23%), lived on Java Island (87.6%), had two children (41.3%), and had no housekeeper to help at their house (62.9%, as shown in Table 1).

Descriptive data for approach coping, avoidance coping, social support, and psychological distress domain scores are shown in Table 2. The results showed that approach coping, avoidance coping, and social support were significantly related to the psychological distress experienced by mothers, and the correlation coefficients were -.284 to .578. Positive associations were found between approach and avoidance coping as well as social support and between avoidance coping with psychological distress. Meanwhile, negative associations were found between avoidance coping with social support and between psychological distress with approach coping and social support.

Moderation effects were analyzed using PROCESS with 5000 bootstrap samples and heteroscedasticity-consistent standard errors with psychological distress as the outcome variable. In Table 3, social support negatively affected psychological distress (B = -.241, 95% CI -.342,.-.142) and served as a moderator in the relationship between approach coping strategies and psychological distress (B = .041, 95% CI .007, -.075). Other moderating effects occurred only when the social support level was moderate (B = .053, 95% CI .014, .091) or high (B = .093, 95% CI .041, .144). In addition, with moderate or high social support as a moderator, the approach coping strategies became insignificant toward psychological distress (B = -.161, 95% CI -.359, .021). See Table 4.

Unlike the approach coping strategies, the inclusion of social support served as a moderator in the relationship between avoidance coping strategies and psychological distress (B = -.100, 95% CI -.138, -.061), as seen in Table 5. The relationship between avoidance coping and psychological distress was significant at one standard deviation below the mean social support score (B = .684, 95% CI .925, 1.336), at the mean social support score (B = .586, 95% CI .542, .628), and at one standard deviation above the mean social support score (B = .487, 95% CI .425, .549). The negative effects of avoidance coping strategies are weaker toward psychological distress when social support moderates. See Table 6.

**Table 1. Demographic and life during pandemic characteristics (N = 1534).**

| Characteristics | Mean | SD |
|---|---|---|
| Age | 37.12 | 6.63 |
| Age of spouse[a] | 39.36 | 7.19 |
| Length of marriage | 11.3 | 6.19 |
| Number of children | 2.11 | 0.94 |
| | **n** | **%** |
| Education | | |
| Junior high school or below | 6 | 0.39 |
| Senior high school | 85 | 5.54 |
| Diploma | 149 | 9.71 |
| University | 1294 | 84.35 |
| Spouse's education | | |
| Junior high school or below | 12 | 0.8 |
| Senior high school | 137 | 9.18 |
| Diploma | 131 | 8.78 |
| University | 1212 | 81.23 |
| Location of residence | | |
| Java Island | 1340 | 87.35 |
| Outside Java Island | 157 | 10.23 |
| Overseas | 37 | 2.41 |
| Working arrangement | | |
| Not working (housewife) | 745 | 48.57 |
| Work from office (WFO) | 80 | 5.21 |
| Work from home (WFH) | 532 | 34.68 |
| Shift (WFO & WFH) | 177 | 11.54 |
| Helped by a housekeeper | | |
| Yes | 569 | 37.09 |
| No | 965 | 62.91 |
| Know someone who is COVID-19 positive | | |
| Yes | 257 | 16.75 |
| No | 1277 | 83.25 |
| Someone in the neighborhood is COVID-19 positive | | |
| Yes | 600 | 39.11 |
| No | 934 | 60.89 |

[a] There were 42 participants who had no spouse (because of divorce or any other reason); therefore, the number of spouses was 1492.

## Discussion

This present study investigated the moderating role of social support on the relationship between coping strategies and psychological distress during the COVID-19 pandemic in Indonesia. We found that approach and avoidance coping strategies showed different interactions with social support and psychological distress. Moreover, social support served as a moderator between the two types of coping and psychological distress among Indonesian mothers are significantly associated.

The results of this study are in line with previous research [25–28], where approach coping strategies and psychological distress were negatively associated. Approach coping strategies consisted of active coping, emotional support, instrumental support, positive reframing,

**Table 2. Descriptive and intercorrelation between approach coping strategy, avoidance coping strategy, social support and psychological distress (N = 1534).**

|  | Mean (SD) | Approach Coping | Avoidance Coping | Social Support | Psychological distress |
|---|---|---|---|---|---|
| Approach Coping | 3.00 (.53) | - | - | - | - |
| Avoidance Coping | 1.84 (.36) | .348 | - | - | - |
| Social Support | 5.48 (.98) | .407 | -.095 | - | - |
| Psychological distress | 0.54 (.39) | -.056 | .578 | -.284 | - |

All correlations are statistically significant; P-value < .05

planning, acceptance, and religious strategies [39]. Moreover, medium and high levels of social support moderated the relationship between approach coping strategies and psychological distress. The results align with previous research where using approach coping strategies and social support significantly correlated with decreased psychological distress [1, 7]. The two implications from these results are (i) the importance of equipping and promoting mothers with various approach coping strategies; (ii) the importance of mothers' significant persons (i.e., spouse, family members, friends, coworkers) to providing an abundance of social support for mothers, specifically in the time of worldwide crises such as the COVID-19 pandemic. Previous studies showed that parental burnout increased during the pandemic [4], and the level of parent stress has not returned to pre-COVID-19 levels [5]. At least a moderate level of social support was needed since low social support did not contribute to decreasing psychological distress [31, 46].

A positive association was also found between avoidance coping strategies and psychological distress, which supported previous studies [29, 30]. Avoidance coping strategies included self-distraction, denial, substance use, behavioral disengagement, venting, self-blame, and humor [39]. At any level, social support can moderate the relationship between avoidance coping strategies and psychological distress. The results were supported by several previous studies where avoidance coping was linked to more anxiety and depression and had a role as a predictor for psychological distress [30, 34]. Social support was identified as a buffering or protective

**Table 3. Social support moderation effect between approach coping strategy and psychological distress.**

| Predictor | β | p | 95%CI | |
|---|---|---|---|---|
|  |  |  | lower | upper |
| Approach Coping | -.161 | .081 | -.359, | .021 |
| Social Support | -.241 | .000* | -.342, | -.142 |
| Approach Coping x Social Support | .041 | .020* | .007, | .075 |

*P-value < .05

**Table 4. Conditional effects of approach coping on psychological distress.**

| Social Support | β | p | 95%CI | |
|---|---|---|---|---|
|  |  |  | lower | upper |
| Low Effect | .013 | .621 | -.038, | .064 |
| Medium Effect | .053 | .007* | .014, | .091 |
| High Effect | .093 | .000* | .041, | .144 |

*P-value < .05

**Table 5. Social support moderation effect between avoidance coping strategy and psychological distress.**

| Predictor | β | p | 95%CI | |
|---|---|---|---|---|
| | | | lower | upper |
| Avoidance Coping | 1.130 | .000* | .925, | 1.336 |
| Social Support | .098 | .010* | .023, | .173 |
| Avoidance Coping x Social Support | -.100 | .000* | -.138, | -.061 |

*P-value < .05

**Table 6. Conditional effects of avoidance coping on psychological distress.**

| Social Support | β | p | 95%CI | |
|---|---|---|---|---|
| | | | lower | upper |
| Low Effect | .684 | .000* | .632, | .736 |
| Medium Effect | .586 | .000* | .542, | .628 |
| High Effect | .487 | .000* | .425, | .549 |

*P-value < .05

source of distress and psychosocial adjustment [47]. The results were also in line with a previous study where people who lack social support were more likely to engage in avoidance coping [31] and the variable was negatively associated with psychological distress [33]. These findings highlighted that even though mothers practice avoidance coping strategies tot increase their psychological distress, the support received from their significant persons were helpful in the buffer of stress experienced.

Several limitations of the present study should be considered. The nature of online data collection might hinder the diversity of samples. Our samples were mostly based in Java Island, and more than half of the participants had a university education background. Thus, the generalization to other sample characteristics should be done cautiously. Future study that reaches wider population groups (e.g., live outside of Java, lower education) is warranted. We acknowledge that the age of the child could potentially be an influential factor to stress anxiety and depression levels. However, in this particular study, we did not collect data on the children's age.

## Conclusion

During the COVID-19 pandemic, social support was negatively associated with psychological distress among mothers in Indonesia. Family, friends, and significant others must support mothers in dealing with various psychological distress. The most important thing for a mother is not the amount and availability of social support, but rather the mother's ability to perceive social support as beneficial for her in this difficult situation.

## Supporting information

**S1 Data.**
(XLSX)

**S1 File.**
(PDF)

## Author Contributions

**Conceptualization:** Fitriani Yustikasari Lubis, Fitri Ariyanti Abidin, Laila Qodariah, Vidya Anindhita, Fredrick Dermawan Purba.

**Data curation:** Fitriani Yustikasari Lubis, Fitri Ariyanti Abidin, Laila Qodariah, Vidya Anindhita, Fredrick Dermawan Purba.

**Formal analysis:** Fitriani Yustikasari Lubis, Fitri Ariyanti Abidin.

**Investigation:** Fitriani Yustikasari Lubis, Fitri Ariyanti Abidin, Vidya Anindhita.

**Methodology:** Fitriani Yustikasari Lubis, Fitri Ariyanti Abidin, Laila Qodariah, Vidya Anindhita, Fredrick Dermawan Purba.

**Project administration:** Fitriani Yustikasari Lubis, Fitri Ariyanti Abidin, Laila Qodariah.

**Supervision:** Fitriani Yustikasari Lubis.

**Writing – original draft:** Fitriani Yustikasari Lubis, Fitri Ariyanti Abidin, Laila Qodariah, Vidya Anindhita, Fredrick Dermawan Purba.

**Writing – review & editing:** Fitriani Yustikasari Lubis, Fitri Ariyanti Abidin, Laila Qodariah, Vidya Anindhita, Fredrick Dermawan Purba.

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
