## [Decision Letter · Decision Letter 0]

6 Jul 2022

PONE-D-21-40890

Coping strategies and psychological distress among mothers during Covid-19 pandemic: the moderating role of social support

PLOS ONE

Dear Dr. Lubis,

Thank you for submitting your manuscript to PLOS ONE. After careful consideration, we feel that it has merit but does not fully meet PLOS ONE’s publication criteria as it currently stands. Therefore, we invite you to submit a revised version of the manuscript that addresses the points raised during the review process.

We look forward to receiving your revised manuscript.

Kind regards,

Silvia Cimino

Academic Editor

PLOS ONE

Reviewers' comments:

Reviewer's Responses to Questions

**Comments to the Author**

1. Is the manuscript technically sound, and do the data support the conclusions?

Reviewer #1: No

2. Has the statistical analysis been performed appropriately and rigorously? 

Reviewer #1: No

3. Have the authors made all data underlying the findings in their manuscript fully available?

Reviewer #1: Yes

4. Is the manuscript presented in an intelligible fashion and written in standard English?

Reviewer #1: No

5. Review Comments to the Author

Reviewer #1: The authors dealt with an important topic; the novelty is that they focused on coping strategies and psychological distress among mothers during the COVID-19 pandemic, with special emphasis on social support.

Abstract; please report participants’ mean age.

Use past tense throughout the entire text, except for the description of measures, where you use present tense.

“The result shows that social support moderates both in the relationship between

approach coping strategy and psychological distress (B=.041, CI .007-.075) and

between avoidant coping strategy and psychological distress (B=-.100, CI -.138--.061).”

Try to being much more specific, such as: “The result shows that HIGHER/LOWER social support moderates both in the relationship between HIGHER/LOWER approach coping strategy and HIGHER/LOWER psychological distress (B=.041, CI .007-.075) and between avoidant HIGHER/LOWER coping strategy and HIGHER/LOWER psychological distress (B=-.100, CI -.138--.061).

“The difference lies in the conditional effect of social support.” This sentence appears erratically presented. Please improve.

“Social support moderates…effect”; this sentence is difficult to understand.

“The results…. distress.” Please revise this part, as these assumptions are difficult to understand.

Introduction; “… greatly impactED the mental….”; as mentioned above, use past tense throughout the text.

“Government responses such as lockdown and quarantine policies might stimulate stress, feeling of uncertainty, anxiety and psychological distress.”; please add references.

“Parents, among others, are also being affected significantly by this pandemic. Parents’ well-being in various aspects of their life: e.g., health, safety, economic might be threaten of these pandemic-related stressors.” Overall, I do kindly but strongly ask you to be much more specific with your claims: ““Parents, among others, are also being affected significantly by this pandemic” But why should this happen?. Parents’ well-being in various aspects of their life: e.g., health, safety, economic might be threaten of these pandemic-related stressors.”; but why should they be threatened? And all really all parents under stress? Did the COVID-19 pandemic negatively impact on all parents, individuals and adults?

“Among these effects of the COVID-19 pandemic toward parenting, specific and more profound effects had been found in the maternal roles.” Please add references; why did the COVID-19 pandemic and its related restrictions impact more severely on mothers? Is it possible that there is a systematic gender-related study bias, such that fathers were not asked, when compared to mothers?

“…since March 2020.”; please add the time lapse.

Social support; please introduce a subheading.

“…. performed by Khalid, A., & Dawood, S. (2020) also revealed… “please double-check referencing style.

Please formulate hypotheses and showcase in much more details, if and to what extent the present data add to the current literature in a new fashion.

Method; while this is often a question of taste and judgement, please start with the description of the participants, then mention the procedure, before you describe the measures.

Participants: inclusion and exclusion criteria: does “aged more than 17 years” equal to “at least 18 years and older”?

SPSS® 21.0 (IBM Corporation, Armonk NY, USA). As regards the use of non-parametric procedures, with such a large sample size, you may use parametric procedures. Please explain, if minimum criteria to run regression models were met.

Results; Tables; never use vertical bars.

Table 5; please report all statistical indices. Please check how you reported the lower interval of the CI.

Table 6 is quite enigmatic. Please explain with more details.

Discussion; compared to the extensive Introduction section, the Discussion section falls very short both by length and content. Please adjust.

Conclusions: strictly taken, it’s not about the amount and availability of social support, but it’s about a person’s ability to perceive social support as beneficial.

6. PLOS authors have the option to publish the peer review history of their article (what does this mean?). If published, this will include your full peer review and any attached files.

Reviewer #1: **Yes: **Serge Brand

---

## [Author Response · Author response to Decision Letter 0]

13 Jan 2023

Submission Coping strategies and psychological distress among mothers during Covid-19 pandemic: the moderating role of social support

Dear reviewer(s), 

First, we would like to thank you for your constructive input and for recognizing our effort to improve this paper. Those comments genuinely boost our motivation to work better on our current study. Regarding your feedback, below we have written down your comments, which we put in italics. Your comments are followed by our response and the associated changes made in the manuscript (identified by track changes). You will find that we adapted the text in all cases. We hope that you will appreciate the new version, and we look forward to finalizing this draft and preparing it for publication. 

Reviewer #1

Overall comment

1. Comments : 

Reviewer #1: The authors dealt with an important topic; the novelty is that they focused on coping strategies and psychological distress among mothers during the COVID-19 pandemic, with special emphasis on social support. 

Our response: 

We thank the reviewer for the constructive comments on this manuscript. 

2. Comments : 

Use past tense throughout the entire text, except for the description of measures, where you use the present tense. 

Our response: 

We checked the whole manuscript and used past tense. 

Abstract

3. Comments : 

Abstract – please report participants’ mean age. 

Our response: 

We added the participants’ mean age and standard deviation

Changes made in the text: 

 Line 9: (Mean age 37.12 years; SD 6.63)

4. Comments : 

Abstract – “The result shows that social support moderates both in the relationship between approach coping strategy and psychological distress (B=.041, CI .007-.075) and

between avoidant coping strategy and psychological distress (B=-.100, CI -.138--.061).”

Try to being much more specific, such as: “The result shows that HIGHER/LOWER social support moderates both in the relationship between HIGHER/LOWER approach coping strategy and HIGHER/LOWER psychological distress (B=.041, CI .007-.075) and between avoidant HIGHER/LOWER coping strategy and HIGHER/LOWER psychological distress (B=-.100, CI -.138--.061).

Our response: 

We thank the reviewer for this comment. 

Changes made in the text: 

Line 12-18: The result shows that moderate and high level of social support moderates the relationship between approach coping strategy and psychological distress (B=.041, CI .007-.075). It means when mother used approach coping, her psychologycal distress mother will decreased further whenever she received moderate and high level of social support. Any level of social support moderates the relationship between avoidant coping and psychological distress (B=-.100, CI -.138--.061). It means when mother used avoidant coping, her social support at any level served as buffer to her psychological distress. 

5. Comments : 

Abstract – “The difference lies in the conditional effect of social support.” This sentence appears erratically presented. Please improve. 

Our response: 

Similar to the previous comment. 

Changes made in the text: 

Line 12-18: The result shows that moderate and high level of social support moderates the relationship between approach coping strategy and psychological distress (B=.041, CI .007-.075). It means when mother used approach coping, her psychologycal distress mother will decreased further whenever she received moderate and high level of social support. Any level of social support moderates the relationship between avoidant coping and psychological distress (B=-.100, CI -.138--.061). It means when mother used avoidant coping, her social support at any level served as buffer to her psychological distress. 

6. Comments : 

Abstract – ““Social support moderates…effect”; this sentence is difficult to understand. 

Our response: 

Similar to the previous comment. 

Changes made in the text: 

Line 12-18: The result shows that moderate and high level of social support moderates the relationship between approach coping strategy and psychological distress (B=.041, CI .007-.075). It means when mother used approach coping, her psychologycal distress mother will decreased further whenever she received moderate and high level of social support. Any level of social support moderates the relationship between avoidant coping and psychological distress (B=-.100, CI -.138--.061). It means when mother used avoidant coping, her social support at any level served as buffer to her psychological distress. 

7. Comments : 

Abstract – “The results…. distress.” Please revise this part, as these assumptions are difficult to understand. 

Our response: 

Similar to the previous comment. 

Changes made in the text: 

Line 12-18: The result shows that moderate and high level of social support moderates the relationship between approach coping strategy and psychological distress (B=.041, CI .007-.075). It means when mother used approach coping, her psychologycal distress mother will decreased further whenever she received moderate and high level of social support. Any level of social support moderates the relationship between avoidant coping and psychological distress (B=-.100, CI -.138--.061). It means when mother used avoidant coping, her social support at any level served as buffer to her psychological distress. . 

Main Manuscript

8. Comments : 

Main Manuscript – Introduction: “… greatly impactED the mental….”; as mentioned above, use past tense throughout the text. 

Our response: 

We checked the whole manuscript and used past tense. 

9. Comments : 

Main Manuscript – Introduction: “Government responses such as lockdown and quarantine policies might stimulate stress, feeling of uncertainty, anxiety and psychological distress.”; please add references. 

Our response: 

We added as suggested. 

Changes made in the text: 

Line 25-29: Government regulations such as lockdown and quarantine policies had an adverse effect on adults’ mental health around the world, as evidenced by an increase in depression and stress-related symptoms [2,3], increased parental burnout [4], and the level of parent’s stress have not yet returned to pre-COVID-19 levels [5].

10. Comments : 

Main Manuscript – Introduction: “Parents, among others, are also being affected significantly by this pandemic. Parents’ well-being in various aspects of their life: e.g., health, safety, economic might be threaten of these pandemic-related stressors.” Overall, I do kindly but strongly ask you to be much more specific with your claims: ““Parents, among others, are also being affected significantly by this pandemic” But why should this happen?. Parents’ well-being in various aspects of their life: e.g., health, safety, economic might be threaten of these pandemic-related stressors.”; but why should they be threatened? And all really all parents under stress? Did the COVID-19 pandemic negatively impact on all parents, individuals and adults? 

Our response: 

We added relevant studies that show that almost everyone was negatively impacted by the Covid-19 Pandemic. 

Changes made in the text: 

Line 29-38: Among others, parents have experienced enormous stress by this pandemic. Due to disruptions in their lives at work, finance, children’s education and in their child care, parents had to deal with a great stressor. [6]. According to a previous study, parents experienced higher levels of stress in many areas of their lives, including work (e.g. unemployment, work-from-home arrangements), family (e.g inability to meet in person, managing children’s education), relationship (conflict with a spouse, lack of support), physical (e.g. poor sleep_, and psychological (e.g. anxiety, depression) [7]. Another previous study found that parental mental health issues are associated with low material assets in a pandemic (such as financial security, food security, and children having access to outdoor space), familial assets (such as parents’ time for themselves), and community assets (receiving support from others outside the household) [8] 

11. Comments : 

Main Manuscript – Introduction: “Among these effects of the COVID-19 pandemic toward parenting, specific and more profound effects had been found in the maternal roles.” Please add references; why did the COVID-19 pandemic and its related restrictions impact more severely on mothers? Is it possible that there is a systematic gender-related study bias, such that fathers were not asked, when compared to mothers? 

Our response: 

We added relevant literature that collects data not only from mothers but also from fathers: Bastiaansen et al. (2021). They found the relationship between state-imposed COVID-19 lockdown measures, co-parenting mitigation, and levels of parental burnout was different between parents: mitigating significant for fathers and non-significant for mothers. It proved the COVID-19 pandemic impacted more severely on mothers.

Changes made in the text: 

Line 43-49 : Among these effects of the COVID-19 pandemic toward parenting, specific and more profound effects had been found in the maternal roles. These effects can be exacerbated in working mothers. According to a previous study, co-parenting had different effect on men and women in pandemic situation [4]. During the pandemic, working mothers spent more time in childrearing and homeschooling than working fathers [11]. The previous study reported that work arrangements during the pandemic resulted in unideal merger of work and personal space of working mothers [12].

12. Comments : 

Main Manuscript – Introduction: “…since March 2020.”; please add the time lapse.

Our response: 

Time-lapse was edit 

Changes made in the text: 

Line 51: since March 2020 until July 2021

13. Comments : 

Main Manuscript – Introduction: Social support; please introduce a subheading. 

Our response: 

We changed the subheading. 

Changes made in the text: 

Line 76 : Social Support

14. Comments : 

Main Manuscript – Introduction: “…. performed by Khalid, A., & Dawood, S. (2020) also revealed… “please double-check referencing style. 

Our response: 

Referencing style was double-checked. 

Changes made in the text: 

Line 89-91 : Social support, self-efficacy, active-practical coping, and active-distractive are negatively associated with psychological distress [30]. 

15. Comments : 

Main Manuscript – Introduction: Please formulate hypotheses and showcase in much more details, if and to what extent the present data add to the current literature in a new fashion. 

Our response: 

We thank the reviewer for the constructive comments to improve this manuscript. 

Changes made in the text: 

-

16. Comments : 

Main Manuscript – Method: while this is often a question of taste and judgement, please start with the description of the participants, then mention the procedure, before you describe the measures. 

Our response: 

As suggested, we moved the participants and procedure sections before the measures section. 

17. Comments : 

Main Manuscript – Method: Participants: inclusion and exclusion criteria: does “aged more than 17 years” equal to “at least 18 years and older”?. 

Our response: 

We revised it as suggested

Changes made in the text: 

Line 146 : at least 18 years and older

18. Comments 1 : 

Main Manuscript – SPSS® 21.0 (IBM Corporation, Armonk NY, USA). As regards the use of non-parametric procedures, with such a large sample size, you may use parametric procedures. Please explain, if minimum criteria to run regression models were met. 

Our response: 

We thank the reviewer for pointing out that we are able to use parametric procedures. We also added a sentence regarding minimum criteria to run regression models, as suggested. 

Changes made in the text: 

Line 204-219 : Descriptive data were presented in mean and standard deviations of variables. Pearson’s correlational analyses were conducted to examine bivariate relationships between psychological distress, social support, and coping strategies. After testing the assumptions, it was found that this data met the minimum criteria for conducting a moderating analysis using regression. Afterward, for testing potential moderator we used moderated regression model and carried out with the PROCESS analysis (v.3) with 5000 bootstrap samples and heteroscedasticity-consistent standard errors [38]. PROCESS is a conditional process modeling program that makes use of an ordinary least squares or logistic-based path analytical framework to test the moderation effect [39]. On the present study, social support is used as a moderator variable (W), psychological distress was the independent variable (X), and approach coping strategies (Y1) and avoidance coping strategies (Y2) were the dependent variable. Two separate model were examine to predicted that (1) Social support would increase the negative correlation between approach coping and psychological distress and (2) Social support would decrease the positive correlation between avoidance coping and psychological distress. The moderator effect for the two-interaction term indicates a significant moderator effect with p < 0.05 and 95% of level confidence interval (CI).

19. Comments : 

Main Manuscript – Results: Tables; never use vertical bars. 

Our response: 

The tables have been revised. 

Changes made in the text: 

Line 229-273: Table 1 to Table 6

20. Comments : 

Main Manuscript – Results: Table 5; please report all statistical indices. Please check how you reported the lower interval of the CI. 

Our response: 

We revised as suggested 

21. Comments : 

Main Manuscript – Results: Table 6 is quite enigmatic. Please explain with more details. 

Our response: 

We clarify the explanation for Table 6. 

22. Comments : 

Main Manuscript – Discussion; compared to the extensive Introduction section, the Discussion section falls very short both by length and content. Please adjust. 

Our response: 

We thank the reviewer for the constructive comments to improve this manuscript. 

23. Comments : 

Main Manuscript – Conclusions: strictly taken, it’s not about the amount and availability of social support, but it’s about a person’s ability to perceive social support as beneficial.

Our response: 

We revised the conclusion as suggested. 

Changes made in the text: 

Line 326-328: Most importantly, it’s not about the amount and availability of social support, but it’s about mother’s ability to perceive social support as beneficial for her in this difficult situation.

---

## [Decision Letter · Decision Letter 1]

17 Jul 2023

PONE-D-21-40890R1Coping strategies and psychological distress among mothers during Covid-19 pandemic: the moderating role of social supportPLOS ONE

Dear Dr. Lubis,

Thank you for submitting your manuscript to PLOS ONE. After careful consideration, we feel that it has merit but does not fully meet PLOS ONE’s publication criteria as it currently stands. Therefore, we invite you to submit a revised version of the manuscript that addresses the points raised during the review process.

We look forward to receiving your revised manuscript.

Kind regards,

Omar M Khraisat, Associate Professor

Academic Editor

PLOS ONE

Journal Requirements:

Reviewers' comments:

Reviewer's Responses to Questions

**Comments to the Author**

1. If the authors have adequately addressed your comments raised in a previous round of review and you feel that this manuscript is now acceptable for publication, you may indicate that here to bypass the “Comments to the Author” section, enter your conflict of interest statement in the “Confidential to Editor” section, and submit your "Accept" recommendation.

Reviewer #2: (No Response)

Reviewer #3: All comments have been addressed

2. Is the manuscript technically sound, and do the data support the conclusions?

Reviewer #2: Yes

Reviewer #3: Yes

3. Has the statistical analysis been performed appropriately and rigorously? 

Reviewer #2: Yes

Reviewer #3: Yes

4. Have the authors made all data underlying the findings in their manuscript fully available?

Reviewer #2: Yes

Reviewer #3: Yes

5. Is the manuscript presented in an intelligible fashion and written in standard English?

Reviewer #2: Yes

Reviewer #3: Yes

6. Review Comments to the Author

Reviewer #2: The authors touch on a very important topic regarding social support, coping and psychological distress and highlighted the need to support mothers in the face of an overwhelming global crisis.

1. The first sentence of your second hypothesis seems to have been a repeat. Kindly check it

2. The last sentence of the “participants” section seems redundant as it is repeated in the next section

3. Under “data analysis" section, authors mentioned that “after testing the assumptions…”, please mention the specific assumptions tested here.

4. Under “data analysis”, please change the word “correlation” to “association” in the sentence that begins with “two separate models…..”

5. Can the authors provide justification why their regression models were not adjusted or controlled for confounders such as age, residence, etc?

6. The authors should also be cautious in their use of “correlated” when in fact they are reporting the results from a regression model. The appropriate word is “associated”.

7. The authors should be cautious with the use of words that connote causal relationships. For instance, in the conclusion section, the authors mention that social support decreases psychological distress. This statement is not factual because the study had a cross-sectional design. The authors should rather say, for instance, “social support is negatively associated with psychological distress”. Kindly check throughout the manuscript to correct these minor errors.

8. Lastly, there are a few grammatical errors that need to be checked to enhance the clarity of the manuscript. I encourage the authors to consider this.

Reviewer #3: Reviewer Comments to the Author

Dear PLOS One team of editorials, thank you for giving me the chance to review the manuscript entitled " Coping Strategies and psychological distress among mothers during COVID-19 Pandemic: the moderating role of social support”

This study gives very important results regarding coping strategies and psychological distress among mothers during the COVID-19 pandemic. However, in a few areas, here are my comments.

1. From the title to the end variation in COVID-19 writing. Better to prefer COVID-19 instead of Covid-19. please correct it

2. While reading the abstract your Respondents are not clear …Which software you used for data analysis is not clear ....it must be included in the abstract…. Make the abstract catchier and reader-friendly...background, methods, results, and conclusion must be clear.

3. In the social support section: your reviewed findings [29,30] were from non-COVID-19 time how could you relate to pandemic time both situations are different …

4. In your sample population mother of neonates, infants, toddlers, preschool, schools, and adolescents were included, and obviously there were lots of variations in the study population, and stress anxiety and depression levels might be affected according to the child’s age ……. how do you justify

5. What are the measures taken to control the confounders?

6. How do you get access to the mother’s network it must be explained. For data collection how do you access the online platform? how and where do you start to share tools first …

7. is convenience sampling a suitable sampling technique for your study ...What about networking or snowball sampling… as you explained your data collection type…please explain why convenience sampling?

8. In line 147: I think the convenience sampling method is for selecting a sample population and an online survey is used for your data collection. A questionnaire (structured or semi-structured, or unstructured) is your tool ...please make it clear

9. In procedure many places future tense is used and needs to be corrected

10. In methods you mention that your present study is part of a longitudinal project …. but in limitation line 318 you mention that the cross-sectional design of this study implies no causation effect can be determined…... So please specify your study design. In your methodology section, you must mention your study design, sampling technique, and research tool …. clearly for ease to the reader.

11. Use correct tense, grammar, sentence, spelling, paraphrase, consistency…etc needs correction

7. PLOS authors have the option to publish the peer review history of their article (what does this mean?). If published, this will include your full peer review and any attached files.

Reviewer #2: **Yes: **Akyirem

Reviewer #3: No

---

## [Author Response · Author response to Decision Letter 1]

30 Aug 2023

Submission Coping strategies and psychological distress among mothers during COVID-19 pandemic: the moderating role of social support

Dear reviewer(s), 

First, we would like to thank you for your constructive input and for recognizing our effort to improve this paper. Those comments genuinely boost our motivation to work better on our current study. Regarding to your feedback, below we have written down your comments, which we put in italics. Your comments are followed by our response and the associated changes made in the manuscript. You will find that we adapted the text in all cases. We hope that you will appreciate the new version and we look forward to finalizing this draft and preparing it for publication. 

 

Reviewer #2

1. Comment: 

From the title to the end variation in COVID-19 writing. Better to prefer COVID-19 instead of Covid-19. please correct it.

Our response: 

Thank you for this remark. We have corrected the COVID-19 writing on the whole manuscript.

Changes made in the text: 

Whole manuscript.

2. Comment: 

While reading the abstract your Respondents are not clear …Which software you used for data analysis is not clear ....it must be included in the abstract…. Make the abstract catchier and reader-friendly...background, methods, results, and conclusion must be clear.

Our response: 

Thank you for this suggestion, we incorporated them in the abstract page 2, especially on line 8-14 for the method explanation.

Changes made in the text: 

Abstract page 2.

3. Comment: 

In the social support section: your reviewed findings [29,30] were from non-COVID-19 time how could you relate to pandemic time both situations are different …

Our response: 

Thank you for your valuable input. We added several findings that shows similar conclusion in COVID-19 time to support our arguments. 

Changes made in the text: 

We have added reference [30] in page 5-6 (line 87-88) highlighted in yellow to support our reviewed findings based on reference [29]. We have added an explanation that the findings in reference [1] were based from a study in the pandemic situation, this also supports our reviewed findings based on reference [31] (previously [30]; see page 5-6 line 93-95, highlighted in yellow).

4. Comment: 

In your sample population mother of neonates, infants, toddlers, preschool, schools, and adolescents were included, and obviously there were lots of variations in the study population, and stress anxiety and depression levels might be affected according to the child’s age ……. how do you justify

Our response: 

We agree with the suggestion(s). We did not collect data on the age of children of the respondents. We added this into the limitation of this study.

Changes made in the text: 

End of discussion section, page 19 line 317-320.

5. Comment: 

What are the measures taken to control the confounders?

Our response: 

Thank you for raising this important point regarding control of the confounders in our study. We did not control the confounders and focused on relationship between the variables that we investigated. 

Changes made in the text: 

No changes have been made.

6. Comment: 

How do you get access to the mother’s network it must be explained. For data collection how do you access the online platform? how and where do you start to share tools first … 

Our response: 

We elaborated the data collection procedures being used in this study.

Changes made in the text: 

We added the explanation in the “procedures” section on page 8-9 line 154-163.

7. Comment: 

Is convenience sampling a suitable sampling technique for your study ...What about networking or snowball sampling… as you explained your data collection type…please explain why convenience sampling? 

Our response: 

We used the term convenience sampling referring to Clark (2017): “A convenience sample is a nonprobability sample that involves a group of elements that is easily accessible to a researcher.”

Reference: Clark, R. (2017). Convenience sample. The Blackwell Encyclopedia of Sociology, 1–2. https://doi.org/10.1002/9781405165518.wbeosc131.pub2

Changes made in the text: 

No changes have been made.

8. Comment: 

In line 147: I think the convenience sampling method is for selecting a sample population and an online survey is used for your data collection. A questionnaire (structured or semi-structured, or unstructured) is your tool ...please make it clear 

Our response: 

Thank you for your valuable input, we incorporate your suggestion in ‘participants’ and ‘procedures’ section.

Changes made in the text: 

Procedure and participants section page 8-9.

9. Comment: 

In procedure many places future tense is used and needs to be corrected 

Our response: 

We thank the reviewer for the constructive comment. We have corrected the tenses in the procedures section.

Changes made in the text: 

We corrected the tenses in procedure section on page 8-9. 

10. Comment: 

In methods you mention that your present study is part of a longitudinal project …. but in limitation line 318 you mention that the cross-sectional design of this study implies no causation effect can be determined…... So please specify your study design. In your methodology section, you must mention your study design, sampling technique, and research tool …. clearly for ease to the reader. 

Our response: 

The present study is part of a longitudinal project investigating the psychological distress of Indonesian mothers and its determinant factors during the Covid-19 pandemic. For the present report, only data from the first data collection time point was presented, using cross-sectional approach.

Changes made in the text: 

We have provided a more detailed explanation of what we mean on methods section on page 8 line 131-144.

11. Comment: 

Use correct tense, grammar, sentence, spelling, paraphrase, consistency…etc needs correction 

Our response: 

We have recheck all manuscript thoroughly.

Changes made in the text: 

Whole manuscript.

Reviewer #3

1. Comment: 

The first sentence of your second hypothesis seems to have been a repeat. Kindly check it

Our response: 

Thank you, we have corrected the second hypothesis.

Changes made in the text: 

Current study section, page 7 line 134-137.

2. Comment: 

The last sentence of the “participants” section seems redundant as it is repeated in the next section

Our response: 

Thank you for your input. We have modified the procedure section to avoid sentence repetition.

Changes made in the text: 

Procedure section, page 8.

3. Comment: 

Under “data analysis" section, authors mentioned that “after testing the assumptions…”, please mention the specific assumptions tested here.

Our response: 

We have elaborate the assumption we tested, namely testing the potential moderator

Changes made in the text: 

Data analysis section, page 11 line 213-220.

4. Comment: 

Under “data analysis”, please change the word “correlation” to “association” in the sentence that begins with “two separate models…..”

Our response: 

Thank you for your input. We have revised it. 

Changes made in the text: 

Data analysis section, page 11 line 217.

5. Comment: 

Can the authors provide justification why their regression models were not adjusted or controlled for confounders such as age, residence, etc?

Our response: 

Thank you for raising this important point regarding control of the confounders in our study. We did not control the confounders and focused on relationship between the variables that we investigated.

Changes made in the text: 

No changes has been made.

6. Comment: 

The authors should also be cautious in their use of “correlated” when in fact they are reporting the results from a regression model. The appropriate word is “associated”.

Our response: 

We have replace the word correlation/correlated into associated/association in the entire manuscript.

Changes made in the text: 

Conclusion section page 19 line 324, discussion section page 18 line 284 and line 300. 

7. Comment: 

The authors should be cautious with the use of words that connote causal relationships. For instance, in the conclusion section, the authors mention that social support decreases psychological distress. This statement is not factual because the study had a cross-sectional design. The authors should rather say, for instance, “social support is negatively associated with psychological distress”. Kindly check throughout the manuscript to correct these minor errors.

Our response: 

We have check throughout the manuscript to correct the words that that connote causal relationships

Changes made in the text: 

Discussion section page 18 line 287 and line 300

8. Comment: 

Lastly, there are a few grammatical errors that need to be checked to enhance the clarity of the manuscript. I encourage the authors to consider this.

Our response: 

We have recheck all manuscript thoroughly.

Changes made in the text: 

Whole document.

---

## [Decision Letter · Decision Letter 2]

27 Oct 2023

PONE-D-21-40890R2Coping strategies and psychological distress among mothers during COVID-19 pandemic: the moderating role of social supportPLOS ONE

Dear Dr. Lubis,

Thank you for submitting your manuscript to PLOS ONE. After careful consideration, we feel that it has merit but does not fully meet PLOS ONE’s publication criteria as it currently stands. Therefore, we invite you to submit a revised version of the manuscript that addresses the points raised during the review process.

We look forward to receiving your revised manuscript.

Kind regards,

Omar M Khraisat, Associate Professor

Academic Editor

PLOS ONE

Journal Requirements:

Reviewers' comments:

Reviewer's Responses to Questions

**Comments to the Author**

1. If the authors have adequately addressed your comments raised in a previous round of review and you feel that this manuscript is now acceptable for publication, you may indicate that here to bypass the “Comments to the Author” section, enter your conflict of interest statement in the “Confidential to Editor” section, and submit your "Accept" recommendation.

Reviewer #3: All comments have been addressed

Reviewer #4: (No Response)

2. Is the manuscript technically sound, and do the data support the conclusions?

Reviewer #3: Yes

Reviewer #4: Yes

3. Has the statistical analysis been performed appropriately and rigorously? 

Reviewer #3: Yes

Reviewer #4: Yes

4. Have the authors made all data underlying the findings in their manuscript fully available?

Reviewer #3: Yes

Reviewer #4: (No Response)

5. Is the manuscript presented in an intelligible fashion and written in standard English?

Reviewer #3: Yes

Reviewer #4: Yes

6. Review Comments to the Author

Reviewer #3: (No Response)

Reviewer #4: Thank you for the opportunity provided peer review for this interesting article on “Coping strategies and psychological distress among mothers during COVID-19 pandemic: the moderating role of social support”. The article involved 1534 Indonesian mothers and it explores that moderate and high levels of social support moderated the relationship between approach coping strategies and psychological distress (B=.041, CI .007-.075). When the mother uses approach coping, her psychological distress will decrease further whenever she receives moderate and high level social support. Any level of social support moderated the relationship between avoidant coping and psychological distress (B=-.100, CI -.138--.061).

1. Online survey was used to collect data and what software was used to enter the data is it the IBM SPSS or other? If you used other data entry software to manage the data centrally then it’s better to indicate.

2. Please provide a reference for the sentence found on line 45 “Among these effects of the COVID-19 pandemic on parenting, a specific and more profound influence had been found in maternal roles.” Since you choose mother as your study population, it needs to be well supported with concrete evidence as mothers are more exposed to stressors than male during the pandemic.

7. PLOS authors have the option to publish the peer review history of their article (what does this mean?). If published, this will include your full peer review and any attached files.

Reviewer #3: No

Reviewer #4: No

---

## [Author Response · Author response to Decision Letter 2]

10 Dec 2023

Submission Coping strategies and psychological distress among mothers during COVID-19 pandemic: the moderating role of social support

Dear reviewer(s), 

First, we would like to thank you for your constructive input and for recognizing our effort to improve this paper. Regarding to your feedback, below we have written down your comments, which we put in italics. Your comments are followed by our response and the associated changes made in the manuscript. You will find that we adapted the text in all cases. We hope that you will appreciate the new version and we look forward to finalizing this draft and preparing it for publication. 

 

Reviewer #4

1. Comment: 

Thank you for the opportunity provided peer review for this interesting article on “Coping strategies and psychological distress among mothers during COVID-19 pandemic: the moderating role of social support”. The article involved 1534 Indonesian mothers and it explores that moderate and high levels of social support moderated the relationship between approach coping strategies and psychological distress (B=.041, CI .007-.075). When the mother uses approach coping, her psychological distress will decrease further whenever she receives moderate and high level social support. Any level of social support moderated the relationship between avoidant coping and psychological distress (B=-.100, CI -.138--.061).

Online survey was used to collect data and what software was used to enter the data is it the IBM SPSS or other? If you used other data entry software to manage the data centrally then it’s better to indicate.

Our response: 

We thank the reviewer for the constructive comment. We have added the software being used in the research. 

Changes made in the text: 

We have added the software information in page 11 (line 220) highlighted in yellow: IBM SPSS 24 software was used to analyze the data.

.

2. Comment: 

 Please provide a reference for the sentence found on line 45 “Among these effects of the COVID-19 pandemic on parenting, a specific and more profound influence had been found in maternal roles.” Since you choose mother as your study population, it needs to be well supported with concrete evidence as mothers are more exposed to stressors than male during the pandemic.

Our response: 

Thank you for your valuable input. We added references that shows mothers are more exposed to stressors than male during the pandemic to support our arguments. 

Changes made in the text: 

We have added reference [11] and [12] in page 3 (line 46) to support our arguments. We also adjust the references number following the additional.

---

## [Decision Letter · Decision Letter 3]

27 Feb 2024

Coping strategies and psychological distress among mothers during COVID-19 pandemic: the moderating role of social support

PONE-D-21-40890R3

Dear Dr.,

We’re pleased to inform you that your manuscript has been judged scientifically suitable for publication and will be formally accepted for publication once it meets all outstanding technical requirements.

Kind regards,

Omar Mohammad Ali Khraisat, Associate Professor

Academic Editor

PLOS ONE

Additional Editor Comments (optional):

Reviewers' comments:

Reviewer's Responses to Questions

**Comments to the Author**

1. If the authors have adequately addressed your comments raised in a previous round of review and you feel that this manuscript is now acceptable for publication, you may indicate that here to bypass the “Comments to the Author” section, enter your conflict of interest statement in the “Confidential to Editor” section, and submit your "Accept" recommendation.

Reviewer #3: All comments have been addressed

2. Is the manuscript technically sound, and do the data support the conclusions?

Reviewer #3: Yes

3. Has the statistical analysis been performed appropriately and rigorously? 

Reviewer #3: Yes

4. Have the authors made all data underlying the findings in their manuscript fully available?

Reviewer #3: Yes

5. Is the manuscript presented in an intelligible fashion and written in standard English?

Reviewer #3: Yes

6. Review Comments to the Author

Reviewer #3: (No Response)

7. PLOS authors have the option to publish the peer review history of their article (what does this mean?). If published, this will include your full peer review and any attached files.

Reviewer #3: No

---

## [Editor Report · Acceptance letter]

23 Mar 2024

PONE-D-21-40890R3 

PLOS ONE

Dear Dr. Lubis, 

I'm pleased to inform you that your manuscript has been deemed suitable for publication in PLOS ONE. Congratulations! Your manuscript is now being handed over to our production team.

Kind regards, 

on behalf of

Dr. Omar Mohammad Ali Khraisat 

Academic Editor

PLOS ONE